# Structural Asymmetry and Kinetic Limping of Single Rotary F-ATP Synthases

**DOI:** 10.3390/molecules24030504

**Published:** 2019-01-30

**Authors:** Hendrik Sielaff, Seiga Yanagisawa, Wayne D. Frasch, Wolfgang Junge, Michael Börsch

**Affiliations:** 1Single-Molecule Microscopy Group, Jena University Hospital, Friedrich Schiller University, 07743 Jena, Germany; michael.boersch@uni-jena.de; 2School of Life Sciences, Arizona State University, Tempe, Arizona, AZ 85287, USA; seiga@asu.edu (S.Y.); frasch@asu.edu (W.D.F.); 3Department of Biology & Chemistry, University of Osnabrück, 49076 Osnabrück, Germany; junge@uos.de

**Keywords:** F_O_F_1_ ATP synthase, *Escherichia coli*, single-molecule fluorescence, symmetry, cryo-EM structure, subunit rotation, elasticity

## Abstract

F-ATP synthases use proton flow through the F_O_ domain to synthesize ATP in the F_1_ domain. In *Escherichia coli*, the enzyme consists of rotor subunits γε**c**_10_ and stator subunits (αβ)_3_δ**ab**_2_. Subunits **c**_10_ or (αβ)_3_ alone are rotationally symmetric. However, symmetry is broken by the **b**_2_ homodimer, which together with subunit δ**a**, forms a single eccentric stalk connecting the membrane embedded F_O_ domain with the soluble F_1_ domain, and the central rotating and curved stalk composed of subunit γε. Although each of the three catalytic binding sites in (αβ)_3_ catalyzes the same set of partial reactions in the time average, they might not be fully equivalent at any moment, because the structural symmetry is broken by contact with **b**_2_δ in F_1_ and with **b**_2_**a** in F_O_. We monitored the enzyme’s rotary progression during ATP hydrolysis by three single-molecule techniques: fluorescence video-microscopy with attached actin filaments, Förster resonance energy transfer between pairs of fluorescence probes, and a polarization assay using gold nanorods. We found that one dwell in the three-stepped rotary progression lasting longer than the other two by a factor of up to 1.6. This effect of the structural asymmetry is small due to the internal elastic coupling.

## 1. Introduction

The F-ATP synthase is a highly flexible and robust motor enzyme [1,2,3,4,5,6,7,8]. ATP synthesis is coupled to the proton motive force (pmf) or ion motive force across the membrane. Under certain physiological conditions, the enzyme can work in reverse and build up an electrochemical potential by hydrolyzing ATP. In *Escherichia coli* (*E. coli*), the F-ATP synthase (EcF_O_F_1_) consists of eight different subunits. The eight subunits either belong to the membrane embedded and proton translocating F_O_ domain (**a**:**b**_2_:**c**_10_), or to the soluble and ATP synthesizing/hydrolyzing F_1_ domain (α_3_:β_3_:γ:δ:ε). Structural details of the EcF_O_ domain were revealed by NMR [9,10], and the structure of the EcF_1_ domain was solved by X-ray crystallography [11]. The structure of the holoenzyme (EcF_O_F_1_) has recently been solved by cryogenic electron microscopy (cryo-EM), and is shown in Figure 1 [12]. The F_1_ domain of this EcF_O_F_1_ structure is similar to the first crystal structure of the F_1_ domain of bovine mitochondrial ATP synthase solved by Abrahams et al. [13] that was followed by a series of structures that showed the F_1_ domain in different conformations [14,15,16,17,18,19,20,21,22]. Recent cryo-EM structures [23] of various ATP synthases [12,24,25,26,27,28,29] revealed the symmetric and asymmetric features of this enzyme. In *E. coli* [12], the main body in the F_1_ domain consists of alternating α and β subunits. Each of the three β subunits at the interface with the adjacent homologous α subunits carries a catalytic nucleotide-binding site, while the respective nucleotide binding site in subunit α is non-catalytic. Each αβ heterodimer forms a catalytic unit, and faces a different side of the central stalk subunit γ. These three αβ heterodimers sequentially cycle through three different conformations, as revealed by the first crystal structure of bovine mitochondrial F_1_ ATP synthase [13]. The (αβ)_3_-ring surrounds the helices of the C-terminal domain (CTD) and the N-terminal domain (NTD) of subunit γ that forms a curved coiled-coil, which asymmetrically interacts with the β subunits. Distal from the N- and C-termini, the globular domain of subunit γ protrudes from the (αβ)_3_-ring, where it docks with the globular NTD of subunit ε and the membrane-embedded ring of **c** subunits of F_O_. Subunits γ, ε, and the **c**-ring form the central rotor stalk. The symmetry of the central stalk is distorted by the eccentric binding of subunit ε and the curved shaft of subunit γ.

The F_O_ and F_1_ domains are also connected by a peripheral, eccentric stalk. In eubacteria, this stalk is composed of a homodimer of **b** subunits that forms a right-handed coiled-coil [30,31,32], which is tethered to the δ subunits, and docks to subunit **a** that serves as the interface with the **c**_10_-ring. Subunit δ binds to the N-termini of all α subunits at the top of the F_1_-head. This rigid stator stalk [33,34,35] is the most obviously asymmetric feature in F-ATP synthases.

The F_O_ domain is located within the inner membrane of bacteria. This bioenergetic coupling membrane separates two phases, one that is acidified and electro-positively charged by the respective proton pump (P-side), and the opposite one that is more alkaline and electro-negative than the former (N-side). Within the F_O_ domain, subunit **a** forms two non-collinear half-channels that connect to either the P-side or the N-side of the membrane, respectively, to provide access for protons and enable protonation of the opposed carboxyl group in each **c** subunit (D61 in *E. coli*), which is located in the middle of the hydrophobic membrane. The number of **c** subunits in the **c**-ring varies between species from 8 to 15 [36,37,38,39,40,41,42,43,44,45,46,47,48,49,50,51,52], but is conserved within species and reflects an adaptation to their environment. In *E. coli*, the ring consists of 10 **c** subunits [53,54,55,56].

Functionally, this enzyme consists of two coupled motors, the multi-step motor in F_O_ (depending on the number of **c** subunits) and the three-step motor in F_1_, which are elastically coupled [5,57,58,59,60]. In the active enzyme, the stator subunits (αβ)_3_δ**ab**_2_ and the rotor subunits γε**c**_10_ rotate against each other. The torque that is generated by the pmf-dependent rotation induced by F_O_ is used to synthesize ATP in F_1_. In reverse, ATP hydrolysis-dependent rotation by F_1_ can pump protons through F_O_ to generate a pmf. The rotation mechanism was coined “binding-change mechanism”, and its principals were first described by Boyer [61,62]. He noticed that the release of ATP, but not its formation, requires energy input. Since then, the mechanism has been confirmed by a series of single molecule rotation experiments, first with the F-ATP synthase [56,63,64,65,66,67,68,69,70,71], and subsequently with the A-ATP synthase and the V-ATPase [72,73,74,75,76,77]. During ATP hydrolysis, the counterclockwise (CCW) rotation (when viewed from the membrane side) of the central stalk (γε**c**_10_) progresses in three major steps of 120°. The constant pauses between steps are characterized by the hydrolysis of ATP and release of Pi, and were coined catalytic dwells [71,78,79,80]. At low ATP concentrations, a second dwell position appeared, thereby dividing the 120° steps into two sub-steps of 80° and 40° in the α_3_β_3_γ-complex of Bacillus PS3 [79]. As its duration depended on the ATP concentration, it was named the ATP binding/waiting dwell, accordingly. This second dwell was also found in other bacterial F-ATP synthases, like EcF_O_F_1_, but not in A/V-type enzymes, where the two dwells are at the same rotor position [73,74,75,76,81]. On the other hand, in human mitochondrial F-ATP synthases, and perhaps in all F-ATP synthases of eukaryotic mitochondrial origin as well, a third sub-step was found and associated with the release of Pi [82]. The details of the rotation mechanism have been reviewed elsewhere [2,3,5,6,7,8,83,84]

The rotation of central stalk subunit γε in the F_1_-head is coupled to the rotation of the **c**-ring in F_O_. During rotation, each **c** subunit is protonated when it passes one of the two half-channels in subunit **a**. Protons from the P-side can enter one half-channel and in *E. coli* are passed to **c**D61 via **a**R210, thereby neutralizing the negatively charged **c**D61. Due to electrostatic constraints, rotation of the **c**-ring is then required to deliver the proton to the other half-channel to complete translocation of the proton across the membrane to the N-side. This alternating protonation/deprotonation of **c** subunits induces a clockwise rotation that is linked to the rotation of the central stalk subunits γ and ε, resulting in ATP synthesis in the catalytic nucleotide-binding sites. In contrast, during ATP hydrolysis, the γ subunit is forced to rotate CCW and protons are pumped in the opposite direction from the N-side to the P-side, accordingly. This protonation and deprotonation of the **c**-ring works like a Brownian ratchet [85,86,87] and provides an almost frictionless rotation of the **c**-ring against the stator to ensure high efficiency and high turnover rates [4].

Similar to the F-type are the A-type ATP synthases of archea and V-type ATPases of vacuoles. Although they differ in structure, the fundamental physical and biochemical principals of ATP synthesis/hydrolysis are the same [88,89]. Crystal structures have revealed that they possess two or three peripheral stalks, respectively, that are formed by a heterodimer and do not enter the membrane and do not contact subunit **a** [90,91,92,93,94,95,96]. Other differences include a collar-like structure formed by subunit C, located perpendicular to the membrane, which serves as an anchor for the peripheral and central stalks and establishes the contact with the **c**-ring [96,97], as well as the central stalk subunits DF. A structural comparison suggests that the coiled-coil domain of subunit D and the globular domain of subunit F are a structural homolog to the γ subunit of F-ATP synthases, while there is no counterpart of the F-type ε subunit [98,99,100]. In addition, subunit F acts as an activator for ATP hydrolysis [101].

In total, the various structural data of F-ATP synthases showed several asymmetric elements that are related to the two stalks. Here, we asked whether structural symmetry breaking caused rotary limping of the active enzyme. In order to correlate structural and rotational asymmetry, we studied actively rotating single EcF_O_F_1_ complexes by three distinct single-molecule fluorescence microscopy techniques.

## 2. Asymmetric Elements in the F-ATP Synthase

The structural data of the F-ATP synthases and rotation experiments with single molecules revealed a fundamental asymmetry in the composition and function of this enzyme. In summary, the static asymmetric structural elements of F-ATP synthases, of which variations can also be found in A-type ATP synthases and V-type ATPases, are as follows: (i) the different conformations and nucleotide occupancies of the three β subunits in the F_1_-head at any moment; (ii) the interface of the **c**-ring with the eccentric subunit **a** including its two half-channels; (iii) the eccentric position of the peripheral stalk consisting of subunits **a**, δ, and **b**_2_; (iv) the interaction between the curved central stalk (especially subunit γ) and each αβ heterodimer at any moment; and (v) the portions of subunits γ and ε that extend beyond the diameter of the **c**-ring might impose a drag on rotation when passing the peripheral stalk. In addition, (vi) in the active enzyme, the catalytic dwell times are different for each dwell position.

### 2.1. The F_1_ Domain

The α and β subunits in the F_1_-head fold the same way with an N-terminal β-barrel domain, an open α/β domain that forms the catalytic site in subunit β, and a C-terminal helical lever domain. The N-terminal β-barrel domain of these subunits forms a crown around the (αβ)_3_-ring that stabilizes the complex as rotation-dependent conformational changes in the β subunits occur. In all structures in all types of ATP synthases, the three pairs of catalytic subunits (subunits αβ in F-type, subunits AB in A/V-type complexes) are present in three asymmetric conformations and have different MgATP/MgADP occupancies, reflecting the various states of catalytic reaction. In addition, each αβ-pair has different interfaces with each of the two other αβ-pairs, as each of the three αβ-pairs is in a different catalytic state, even in the absence of the central rotor stalk [102]. This is consistent with the observation that the three catalytic sites bind nucleotides with low, medium, and high affinity only in the presence of Mg^2+^ [103]. It appears that binding of Mg-nucleotides induces conformational changes in the enzyme that are required for its activity. The asymmetric subunits were coined according to their nucleotide occupancy in the first crystal structure from bovine mitochondrial F_1_-ATP synthase [13], namely α/β_TP_ for the site with a bound AMP-PNP (a non-hydrolysable ATP analog) with high binding affinity, α/β_DP_ for the site with a bound ADP and medium binding affinity, and α/β_E_ for an empty and open site with low binding affinity [104]. During catalysis, the three αβ heterodimers unidirectionally cycle through all three conformations with transition states that are represented by different crystal structures. The conformations of the β subunits are coordinated by their position relative to the curved γ subunit via electrostatic interactions. During this process, not only are single domains of subunits moving, but the entire F_1_-head is rocking [12]. In total, although the chemical reactions in each nucleotide-binding site are symmetric in the time average, the overall structure of the F_1_-head is asymmetric at any particular moment.

### 2.2. The F_O_ Domain

In the F_O_ domain, proton flux across the membrane is realized by the ring of **c** subunits and the peripheral subunit **a** that provides two half-channels for protons to access the **c**-ring. In *E. coli*, each of the ten **c** subunits consists of a hairpin structure with two α-helices that spans the membrane, with an essential D61 that can bind a proton. Together they form a symmetric **c**-ring. At the interface with subunit **a**, the **c**-ring faces the two half-channels, with the essential residue R210 in subunit **a** that mediates proton transport from the medium to subunit **c**. Although the interface is located asymmetrically on one side, each c subunit is exposed to the same conditions when passing the interface, therefore rotational symmetry is maintained. However, as the central stalk subunits γ and ε bind asymmetrically to the **c**-ring, each **c** subunit is unique. Neither is the **c**-ring rotating symmetrically in synthesis and hydrolysis directions, as the two α-helices of each **c** subunit are different.

Cryo-EM structures from different organisms revealed that the two transmembrane helices that comprise the half-channels in subunit **a** cross the membrane at an oblique angle to cover one site of the **c**-ring [105]. When viewed from the membrane side, the two half-channels are asymmetrically arranged: the right semi-channel turns towards the P-side of the membrane, while the left semi-channel faces the N-side, to ensure obliged proton translocation into one direction only that is linked to the rotation of the central stalk and the catalysis reaction in F_1_ [106,107]. Recent experiments have shown that access to the two half-channels is not the same from both sides of the membrane [108].

The ratio of **c** subunits or translocated protons to αβ heterodimers or catalyzed nucleotides, respectively, is not an integral number. In EcF_O_F_1_, the ratio is 10/3, i.e., 3.3 protons are translocated to synthesize one ATP molecule. In a rigid coupling of the two motors, a varying number of 3 to 4 protons would be translocated during ATP synthesis that would be the basis for an asymmetric rotation. However, the **c**-ring is elastically coupled to the central shaft in F_1_ [58] to provide a smooth rotation [57].

### 2.3. The Peripheral Stalk

F-ATP synthases have one eccentric peripheral stalk that covers almost the whole complex, i.e., a distance of more than 10 nm. This stalk is the most prominent asymmetric feature of the F-ATP synthase. The **b**_2_ homodimer that forms the main body of this stalk is an intrinsic asymmetric structure, as each individual **b** subunit forms different contacts with other subunits. The helix of one monomer (**b**_N_) that is skewed toward the N-terminus forms contacts with subunit δ at the top and with the N-terminus of one subunit α on the side of the F_1_-head. The helix of the other monomer (**b**_C_) is skewed towards the C-terminus and was found in close proximity to one β subunit. The **b**_2_ homodimer interacts with the αβ heterodimer at a location close to the non-catalytic nucleotide-binding site [109,110], such that it is unlikely to impact nucleotide binding or release at the catalytic site. The **b**_2_ homodimer bifurcates just above the membrane to dock to subunit **a** at two locations [12]. On the other side, subunit δ docks the peripheral stalk via one **b** subunit to the N-termini of the three α subunits. However, the α-δ contact region is different for each α subunit [12], and the αβ-pair that faces **b**_2_δ is therefore different than the other two. These structural features all contribute to the asymmetrical nature of the peripheral stalk.

In contrast, A-type and V-type complexes have two and three peripheral stalks, respectively, formed by heterodimers that are structural homologues of the F-type stalk [90,91,92]. The stalks are asymmetric, as each stalk contacts different subunits and adopts a unique bend that depends on the rotational state of the complex. The two stalks of A-ATP synthases are not directly opposite to each other, but are on each side of one AB-pair [96]. In contrast, the three peripheral stalks in V-ATPases make nearly indistinguishable contacts with the V_1_ head, but have different contacts with the collar-like structure [111]. As the ration of AB-pairs to peripheral stalks is 1:1 in V-ATPases, it would be interesting to investigate if this is reflected in a different kind of asymmetry in the rotation pattern of the central stalk subunit DF compared to that of F-ATP synthases.

### 2.4. The Central Stalk

The central stalk consists of subunits γ and ε in the F_1_ and the **c**-ring in the F_O_ domain. It was revealed by the first crystal structure of the bovine mitochondrial F_1_ domain [13] that the curved shaft formed by the coiled-coil double helix of subunit γ interacts at two locations with the β subunits. First, within the cavity of the F_1_-head, residues γR268 and γQ269 form a “catch” with a loop in subunit β_E_ that encompasses residues β297–305. This catch-loop results from salt bridges and H-bond interactions and is important for the activity of the complex [112]. Second, residues γ80–90 form a bulge near the orifice of the central cavity of F_1_ that interact with residues β394-400 (DELSEED-loop) at the end of the lever-domain of subunit β_TP_ [13]. In addition, the top of subunit γ forms a swivel joint that can unwind during ATP hydrolysis [113]. On the F_O_ side, the globular domains of subunits γ and ε form an asymmetric interface with the **c**-ring that breaks its intrinsic symmetry. These elements provide asymmetry to the central shaft that connects the biochemical events in the nucleotide-binding sites to the proton flow in the F_O_ domain, and are important for unidirectional rotation with high kinetic efficiency, although slow hydrolytic reactions can occur also in the shaft-less (αβ)_3_-ring [102,114].

The regulatory subunit ε, which docks to the globular domain of subunit γ, comprises a β-barrel at the NTD, and a helix-turn-helix motive at the CTD. The NTD is located eccentrically on top of the membrane bound **c**-ring, where it rotates together with the globular domain of subunit γ. The CTD of subunit ε can extend into the (αβ)_3_-ring of F_1_ and form several contacts with subunits α, β, and γ [11]. In this extended conformation, it acts as an intrinsic inhibitor of ATP hydrolysis. The mechanism of subunit ε inhibition has been reviewed in detail elsewhere [115,116,117].

During rotation, subunit ε comes close to the **b**_2_ homodimer of the peripheral stalk once every full turn. However, as there is no cryo-EM or crystal structure available when subunit ε and the **b**_2_ homodimer are in closest proximity, it is unknown if the peripheral stalk imposes any steric hindrance on the rotation of subunit ε. This question can be answered only by single molecule experiments of the actively rotating F-ATP synthase. Any strong interaction would be revealed by one extended dwell or rotational step, with reduced velocity. In A-ATP synthases, two such events would be expected.

### 2.5. Rotational Catalysis

The rotary mechanism of the F-ATP synthase during ATP hydrolysis follows an alternating site binding-change mechanism [62]. At concentrations of ATP that saturate the rate of ATP hydrolysis activity ([ATP] > k_M_), the ATP hydrolysis-dependent rotation occurs in 120° power strokes [58,118,119] that occur on a µs time scale [120], and are interrupted by catalytic dwells with a duration on the order of a few ms [78]. During each catalytic dwell of eubacterial F-ATP synthases, ATP is hydrolyzed at the catalytic site that corresponds to β_DP_ [22], and the dwell ends upon release of Pi to create an empty site β_E_ [70,71,79]. Rotation of subunit γ during the ensuing 120° power stroke undergoes a series of accelerations and decelerations that have been divided into two 60° phases [60,121]. The binding of ATP to the empty catalytic site occurs during phase-2 [60].

When the ATP concentration becomes rate-limiting ([ATP] < k_M_), an ATP-binding dwell is observed that interrupts the power stroke at some point during phase-1, and occurs most frequently 34°–40° after the catalytic dwell [60,78]. The duration of ATP-binding dwells vary inversely with ATP concentration. Although the power stroke of the A-ATP synthase is closely similar to that of F-ATP synthases, ATP-binding dwells are not observed at rate-limiting ATP concentrations, such that ATP binding is believed to occur during the catalytic dwell [73,76,122]. The dissociation of ADP from β_DP_ typically occurs during phase-2 [71,121]. As the power stroke ends, the conformations of the catalytic sites change, such that β_E_ changes to β_TP_, β_TP_ changes to β_DP_, and β_DP_ changes to β_E_.

As a result of each power stroke, subunit γ rotates by an amount equivalent to one αβ heterodimer in the (αβ)_3_-ring, and involves the net binding of one ATP and release of one ADP and one Pi. Consequently, each complete rotation of subunit γ requires the consumption of an ATP at each catalytic site in the (αβ)_3_-ring for a total of three ATP. This alternating site mechanism is supported by structural evidence [12,13,14,15,16,17,18,19,20,21,22], by nucleotide-binding studies [123,124,125,126], and by single-molecule studies that have visualized rotation directly using visible probes attached to the rotor. These probes include actin filaments [58,63,118,127], gold/polystyrene/magnetic beads [70,71,78,79], and gold nanorods [60,76,120,121,128], as well as single-molecule Förster resonance energy transfer (smFRET) measurements [55,59,67,68,69,129,130,131,132,133,134].

## 3. Single Molecule Rotation Experiments

The first rotation experiments with the F_1_ domain from the thermophilic *Bacillus* PS3 were described in 1997 by Noji et al. [63]. They immobilized the (αβ)_3_-ring on a glass surface, attached a fluorescent actin filament to subunit γ and observed its rotation after addition of ATP. Since then, a variety of rotation experiments with F_1_ or F_O_F_1_ from different organisms have been performed. In addition to actin filaments, polystyrene beads, gold beads, magnetic beads, and gold nanorods have been used as a probe for rotation. Recently, the rotation of A/V-type complexes have also been examined. Here, we focus on the *E. coli* F-ATP synthase to investigate the rotary asymmetry in the holoenzyme. We have reevaluated our previously published data [58,108,119,130,133] to search for persistent asymmetric patterns. In each study, the use of a complete EcF_O_F_1_ complex was confirmed by SDS-PAGE [56,130,135], and by the rotation experiments that were designed such that rotation was only visible in the presence of the F_1_ and the F_O_ domain. ATP hydrolysis driven rotation of the rotor subunits was observed via an attached fluorescent actin filament or gold nanorod (AuNR) to the **c**-ring, or via smFRET between donor and acceptor fluorophores on individual stator and rotor subunits (Figure 2). ATP concentrations above the k_M_ of 45 µM for *E. coli* [136] ensured continuous rotation in 120° steps with catalytic dwells in between.

### 3.1. Rotation Experiments with Actin Filaments

For the kinetic analysis of rotating EcF_O_F_1_ with actin filaments, we have reevaluated some previously unpublished data and data published by Sielaff et al. [58,119]. Single molecules of EcF_O_F_1_ were immobilized via His-tags in each β subunit on a Ni-NTA covered glass surface. A fluorescent actin filament (length: 0.5–0.9 µm) coupled to the **c**-ring via a Strep-tag served as a reporter to observe the ATP (0.05–5 mM) hydrolysis driven CCW rotation of the central stalk via micro-videography (Figure 2a). The rotational rate and dwell times were independent of the filament length and ATP, respectively (Figure 3a). The progression of a stepwise rotating filament is shown in Figure 3b. Histograms of the angular dependent probability distribution with three peaks representing the three catalytic dwells were fitted with Gaussians to determine the center of the peak (Figure 3c), whereas the mean dwell time of the complex at each of the three positions was determined with a monoexponential fit (Figure 3d). The propensity to reside in any out of the three positions differed, resulting in varying peak highs and dwell times. As the absolute orientation of the enzyme could not be determined, we estimated the maximum possible asymmetry between peaks and dwell times by aligning the peaks with the highest dwell time of each complex (Figure 3e). This resulted in one large peak, which area was 55–59% larger than that of the two other peaks. The averaged dwell times (τ) of the three peaks were 0.3 ± 0.2 s, 0.2 ± 0.15 s, and 0.2 ± 0.12 s, i.e., one long and two short dwells. The dwell times show a high standard deviation due to the fast rotational velocity and the short recording time of some filaments. The dwell times of the largest peak were on average 1.5 times longer than the other two. These results indicate a rotational asymmetry of the complex. However, the position of the peripheral stalk in relation to the orientation of the central stalk remained unknown. Therefore, these results are only supportive to the following experiments.

### 3.2. Rotation Experiments with smFRET

In another set of single-molecule fluorescence measurements, smFRET was applied to investigate subunit rotation in liposome-reconstituted EcF_O_F_1_ during ATP hydrolysis and ATP synthesis [55,59,67,68,69,130,131,133,134]. To monitor the rotary motion of subunit ε during catalysis, the FRET donor fluorophore tetramethylrhodamine–C_5_–maleimide was bound to a cysteine in subunit ε (εH56C) and the FRET acceptor fluorophore Cy5bis–C_5_–maleimide was coupled to crosslink the two **b** subunits of the peripheral stalk (**b**Q64C). Single molecules of EcF_O_F_1_ were observed in freely diffusing liposomes using a confocal microscope setup (Figure 2b). The rotation of the rotor subunit ε relative to the static **b**_2_ homodimer was recorded for ATP-driven ATP hydrolysis at saturating ATP concentrations (1 mM) and for proton-driven ATP synthesis at a high proton motive force, i.e., comprising an initial proton concentration difference of 4.1 pH units and an electric membrane potential. Excitation of the donor fluorophore resulted in photon bursts with different donor and acceptor intensities. The observation time of the freely diffusing proteoliposomes was limited to a few hundred milliseconds, and one, two, or three FRET efficiency levels could be discriminated in each photon burst. As before, these FRET levels corresponded to the catalytic dwells in between power strokes. The length of each dwell for the three distinct FRET levels was plotted in a histogram in order to calculate the mean dwell time for each FRET level separately. For the kinetic analysis, the first and last FRET level of a photon burst were omitted, as their actual duration before entering and after leaving the confocal detection volume remained unknown. For both ATP hydrolysis and ATP synthesis, three FRET levels were observed representing the three distances between the FRET labels or the relative orientation of the subunits, respectively, during the 120° rotation in EcF_O_F_1_. Average dwell times for the three orientations were 12.7 ± 1.0 ms (medium FRET efficiency, M-level), 17.6 ± 1.4 ms (low FRET efficiency, L-level), and 15.8 ± 1.7 ms (high FRET efficiency, H-level) during ATP hydrolysis (Figure 4a), and 15.4 ± 1.0 ms (M-level), 24.0 ± 4.4 ms (L-level), and 19.5 ± 2.2 ms (H-level) for ATP synthesis (Figure 4b). In both cases, kinetic analysis yielded an asymmetric distribution of FRET level dwells. The M-level had a significantly shorter dwell time than the l- and H-level, and the short and long dwell times differed by factors of 1.39 and 1.56 for ATP hydrolysis and synthesis, respectively [130].

We also compared the occurrence of each FRET level in the dwell time histograms and found a higher occurrence of the M-level based on photon bursts with three or four FRET levels. This is in line with the shorter dwell time of the M-level and was confirmed by Monte-Carlo simulations, in which the abundance of each FRET level with the experimentally determined dwell times during ATP hydrolysis was examined using the experimentally determined dwell times for each orientation [131]. In case of three-level photon bursts, the following FRET sequences were found in the experiment/simulations, respectively: H-M-L 63.3%/38.4%, M-L-H 23.3%/33.6%, and L-H-M 13.3%/28.0%. The simulations with 10000 bursts and 1000 repetitions were statistically consistent and qualitatively reproduced the asymmetry in the experimental FRET level distribution with a strongly preferred intermediary M-level. However, the experimental data sets were limited to less than 300 molecules or photon bursts, respectively.

In subsequent smFRET experiments, the FRET donor EGFP was fused to the C-terminus of subunit **a** and the acceptor fluorophore Alexa568 to subunit ε at position ε56 [133]. Dwell times of 9 ± 1 ms (L-level), 11 ± 1 ms (M-level), and 10 ± 1 ms (H-level) were observed during ATP hydrolysis. All dwells were similar within the error margin. When the FRET acceptor was coupled to subunit γ at position γ106 instead, the dwell times changed to 20 ± 3 ms (L-level), 20 ± 1 ms (M-level), and 16 ± 1 ms (H-level). Here, the dwell time of the H-level was shorter than the other two, but overall the difference between them was less than a factor of 1.25. The total number of photon bursts analyzed for dwell times in this study was similarly low, i.e., less than 400 enzymes for smFRET between subunits **a** and ε.

### 3.3. Rotation Experiments with Gold Nanorods

In this assay, instead of an actin filament, an 80 × 40 nm AuNR was attached to the **c**-ring of EcF_O_F_1_ as a visible probe, and ATP hydrolysis driven rotation of EcF_O_F_1_ was measured. The ability to acquire data at speeds equivalent to 100 k frames per s in combination with an AuNR substantially improved the time resolution to 10 µs, and increased the accuracy of determining rotational position to a standard error of 0.02°–0.12° [56]. When passed through a polarizing filter and measured by an avalanche photo diode, the polarized red light scattered from an AuNR of these dimensions changed in a sinusoidal manner as a function of the rotary position of the AuNR versus the polarizer direction. For each dataset, the polarizer was aligned with each AuNR-EcF_O_F_1_ complex to minimize the scattered red light intensity during one of the three catalytic dwells. The light intensity data collected from the subsequent 120° power stroke resulted in an increase from a minimum through a maximum, at which point the AuNR has rotated 90° [56]. These power strokes were selected for further analysis from the total of three successive power strokes that are required to complete 360° of rotation (Figure 5a).

After EcF_O_F_1_ embedded lipid bilayer nanodiscs (n-EcF_O_F_1_) were attached to a glass slide via His_6_-tags on the N-termini of the β subunits, ATP hydrolysis-dependent rotation of an AuNR attached to the **c**_10_-ring of n-EcF_O_F_1_ was examined in the presence of saturating concentrations of ATP (1 mM MgATP) at different pH. Under these conditions, transient dwells were observed to occur at ~36° intervals between the catalytic dwells of the ATP hydrolysis-dependent CCW power stroke. The transient dwells were shown to result from an interaction in EcF_O_ between subunit **a** and successive **c** subunits in the **c**_10_-ring, which is consistent with their occurrence every 36°. This interaction was not only able to cancel the force of the ATP hydrolysis-driven CCW **c**-ring rotation, but over 70% of the time was able to push the rotor backwards in CW (i.e., ATP synthesis) direction by as much as one **c** subunit [137]. The average duration of transient dwells ranged from 50 µs to 175 µs [56], which is on the same time scale as proton gradient-dependent rotational stepping of **c** subunits during ATP synthesis under typical in vivo conditions (Figure 5b).

Each rotational data set analyzed from an n-EcF_O_F_1_ complex contained ~300 power strokes that increased from a minimum through a maximum intensity, which were examined for the presence of transient dwells. Between pH 9.0 and 7.5, about 23% of the power strokes analyzed from each data set contained transient dwells (Figure 5b). As the pH decreased below pH 7.5, the percentage of power strokes containing transient dwells increased inversely with pH to a maximum average of 50% at pH 5.0 (Figure 5d) [108]. For any given molecule, there is an equal probability that the power stroke analyzed corresponds to the rotation that originates from one of the three structural states (see below). This was corroborated by the fit of the distribution in Figure 5c with three Gaussian curves. The probability of forming transient dwells in each dataset was either low (blue line), medium (red line), or high (green line). The measured distribution at each pH (gray bars) was well fitted by the sum of the three Gaussians (black line).

The pH dependence of transient dwell formation observed with n-EcF_O_F_1_ was comparable to the pH range that powers proton gradient-driven ATP synthesis that occurs when the cytoplasmic and periplasmic half-channels in subunit **a** are exposed to pH 8.5 and 5.5, respectively. However, when EcF_O_ is embedded in a nanodisc, both proton half-channels are exposed to the same pH. The pH dependence of transient dwell formation shows that the periplasmic half-channel is more easily protonated in a manner that halts ATP hydrolysis-driven rotation by blocking ATP hydrolysis-dependent proton pumping. The distribution of the percentage of transient dwells formed per data set did not fit to a single Gaussian at any of the pH values examined. In several data sets at pH values between 5.0 and 6.0, transient dwells were observed in 90–100% of the power strokes. The fits of the distributions of power strokes containing transient dwells improved substantially when the sum of three Gaussian distributions was used (Figure 5c). Each of the three Gaussians represents the ability to form transient dwells during the ATP hydrolysis driven power stroke, which occurred with a high, medium, or low probability, respectively. The overall formation of transient dwells increased with an increase in acidic environment (Figure 5d). For example, when the data were fit in this manner at pH 5.0, the high, medium, and low probabilities averaged 70%, 53%, and 29% of transient dwells formed per data set, respectively.

## 4. Discussion

In *E. coli* F-ATP synthase, rotation of the central shaft subunits γ and ε progresses in steps of 120° during ATP hydrolysis. At high (mM) ATP concentrations, the dwells of the rotor between steps are associated with the hydrolysis reaction of ATP. Here, we compared the dwell times of single enzymes measured by three different single-molecule techniques and found asymmetric rotational behavior with all three approaches. In experiments with actin filaments and large beads, it could be argued that an apparent rotational asymmetry originated from the drag of the probe or its interaction with the surface. However, asymmetric dwell times were also present when these surface effects were minimized in single-molecule experiments with nanorods, or even completely absent in smFRET measurements in solution.

### 4.1. Asymmetry Corroborated from Single-Molecule Rotation Experiments

In the rotation assay with actin filaments, we found one long (300 ms) and two short dwell times (200 ms). The areas of the Gaussians associated with these dwells differed by a factor of about 2, while the dwell times differed by a factor of 1.5. However, the significance of these findings is limited, because the rotary movement is damped by the viscous drag of the actin filament and the uncertainty of the angular position of the filament due to its inherent flexibility. The existence of one longer dwell time seemed to be plausible in view of the peripheral stalk subunits **b**_2_ that might affect the conformational dynamics of the nearest catalytic binding site or the progression of the central shaft when the bulge of subunits γ or ε pass by. In contrast, in smFRET experiments, the rotation is not hampered by a viscous drag of the marker of rotation, and the F-ATP synthase can rotate at maximum speed under non-limiting ATP concentrations. Different dwell times and ratios were measured depending on the labeling positions for the FRET donor and acceptor fluorophores. However, the dwell times differed not more than by a factor of 1.56, which is in the same order of magnitude as for the rotation assay with actin filament.

Other research groups also reported on asymmetric rotation with different types of ATP synthases in single-molecule experiments with immobilized complexes. Similar to the rotation experiments described here, small gold beads (40–80 nm) [138,139] or magnetic beads (200 nm) [140] were attached to immobilized F-type ATP synthases to reduce the viscous drag of the probe. The rotation of the central stalk subunits, DF in A/V-type complexes of *Thermus thermophiles* (TtA_O_A_1_) [74] and *Enterococcus hirae* [141] was investigated with 40 nm gold beads. In all studies, the histograms of angular orientation showed peaks with different heights and widths, similar to data shown here in Figure 3. From these data we draw two conclusions. From the broad width of the peaks, we deduce that the rotation of the central shaft was not always perfectly C3-symmetric. Instead, individual steps could be larger or smaller than 120°, although one full rotation was always 360°. The different heights indicate that the catalytic dwell times vary from each for the three positions. The variety of these data confirms that asymmetric rotation is not an artifact but a common feature of this flexible enzyme.

### 4.2. Comparison of Cryo-EM Structures

Structural evidence for the asymmetric distribution of rotor orientations with respect to the peripheral stator stalk has been provided recently by several cryo-EM structures of distinct ATP synthases. These studies resolved three different conformations of the F-type ATP synthase with the central shaft rotated by approximately 120° in each of them. The structures were classified according to the orientation of the central stalk, i.e., the bulge of subunit γ in relation to the peripheral stalk. The three structural states appeared as different subpopulations of the enzymes on the cryo-EM grid. For EcF_O_F_1_, Sobti et al. [12] reported a ratio for the three rotor orientations of 46.3%, 30%, and 23.7% for state 1, state 2, and state 3, respectively (Figure 6). Different subpopulations and ratios were also found for the *Bacillus* PS3 F-ATP synthase (BPF_O_F_1_) [29], the spinach chloroplast enzyme (CF_O_F_1_) [27], the yeast vacuolar V-ATPase (YV_O_V_1_) [94], and TtA_O_A_1_ [96]. However, almost no differences were found for the subpopulations of the three states of the bovine mitochondrial F-ATP synthase (MF_O_F_1_) [24]. Table 1 summarizes the occurrence of the three rotor orientations for the different enzymes. The relative ratios are similar for the two bacterial F-ATP synthases, with state 1 being the most abundant. In contrast, in CF_O_F_1_ the most abundant rotor orientation was found >10 times more often in state 3 than in the other two states. In addition, the rotary angles of the central stalk between the three states were 103°, 112°, and 145°, which deviates significantly from the expected 120° angles. As TtA_O_A_1_ and YV_O_V_1_ have two and three peripheral stalks, respectively, their structures cannot be aligned with the F-type ATP synthases. Still, the ratios of the three states are similar to the ones of CF_O_F_1_ and BPF_O_F_1_, respectively.

### 4.3. Correlation of Cryo-EM Structures with smFRET Data

The questions remains of how the different dwell times found in single-molecule FRET experiments can be related to the three structures of EcF_O_F_1_ (Figure 6). FRET efficiencies for EcF_O_F_1_ labeled at rotor subunit ε56 and stator subunit **b**64 corresponded to fluorophore distances of 8.3 nm (L*-level), 6.4 nm (M*-level), and 4.0 nm (H*-level) in the absence of any added nucleotides, or in the presence of 1 mM ADP and/or 3 mM Pi. In the presence of 1 mM ATP during catalysis, the respective distances were 7.8 nm (L-level), 6.3 nm (M-level), and 4.6 nm (H-level) [130]. In the three EcF_O_F_1_ cryo-EM structures [12], distances between the Cα atoms of these two residues were 8.2/8.6 nm (PDB-id: 5T4P, state 2), 6.1/7.2 nm (PDB-id: 5P4O, state 1), and 3.5/3.4 nm (PDB-id: 4T4Q, state 3), respectively, depending on the position of **b**64 in each **b** subunit (Figure 6). The cryo-EM structures do contain only one bound ADP in one closed catalytic binding site, while the other two sites are empty and open. Therefore, the apparent agreement with the smFRET data in the absence of nucleotides or in the presence of ADP/Pi is excellent and approaches the current “state-of-the-art” of single-molecule FRET-based precise distance calculations [142]. However, given the uncertainties of the additional linker length for the fluorophores, limited mobility of the fluorophores in their local environment on the enzyme, and errors in fluorescence quantum yield determination in our smFRET data, the FRET distances obtained during catalysis and subunit rotation are also similar to the cryo-EM enzyme structures trapped in non-catalytic states.

The cryo-EM structures were based on more than 10,000 enzymes for each state [12,24]. The population of states was 46% for state 1, 30% for state 2, and 24% for state 3 (Table 1). They correspond to the medium (M*- or M-level), low (L*- or L-level), and high (H*- or H-level) FRET efficiencies, respectively (Figure 6). In contrast, in smFRET data in the absence of nucleotides or in the presence of ADP/Pi, the relative number of M*-level found was only 15%. Most of the non-active enzymes were found with the rotor orientation associated with the L*-level (50%). The H*-level occurred in 35% of cases (Figure 4c) [130]. About 1000 enzymes contributed to each smFRET data set, i.e., significantly less than in the cryo-EM data set. The catalytically active EcF_O_F_1_ showed a nearly uniform distribution of L, M, and H-level, based on only a few hundred enzymes. Thus, the asymmetric distributions found in cryo-EM structures and in smFRET data do not match.

This might be explained in part by the constraints of the smFRET analysis using freely diffusing proteoliposomes, but also by a potential bias of particles that can adsorb to the cryo-EM grid in a preferred orientation [143,144]. The relative occurrence of the L-, M-, and H-level during ATP hydrolysis is correlated to the dwell time of the respective FRET level [131]. To add a FRET level to the dwell time distribution requires a photon burst with at least three FRET levels, because the first and last FRET level cannot be included. The diffusion time of proteoliposomes through the confocal detection volume, i.e., the time for the occurrence of three consecutive FRET levels in a photon burst, was less than 1 s. The minimal dwell time that could be assigned in the smFRET time traces was about 5 ms, and the shortest photon burst analyzed was 20 ms. Therefore, to be added to the dwell time histograms, photon bursts had to contain short dwells as intermediate FRET levels. For both ATP hydrolysis and ATP synthesis conditions, the dwell times of the M-level were shorter than for the L- and H-level. Accordingly, the M-level were most abundant in the dwell time distributions. However, Monte-Carlo simulations for the relation of dwell time and occurrence of a FRET level revealed that more than 10,000 FRET photon bursts with three and more subsequent FRET levels are required for a compelling correlation between them. In the smFRET experiments, dwell times had to be analyzed based on less than 100 dwells per FRET level.

Alternatively, the observation times of proteoliposomes in solution have to be prolonged to obtain more consecutive FRET levels within one photon burst. One approach to extend observation times in confocal smFRET measurements is the “anti-Brownian electrokinetic trap” (ABEL trap) invented by Cohen and Moerner [145,146,147,148,149,150,151,152,153,154,155]. A confocal laser pattern with correlated and localized photon detection is used to determine the position of a fluorescent particle within thin microfluidics. By applying electric fields, the fluorescent molecule is pushed back to the center of the laser focus pattern. In preliminary experiments, FRET-labeled EcF_O_F_1_ could be held and analyzed for up to 4 s in the absence of ATP until photobleaching of the dyes [156,157,158,159]. The limiting 1 µm thickness of the microfluidics and the uniform illumination by a fast moving laser focus yields constant fluorescence intensity from a single trapped molecule. Well-known smFRET artefacts, like red-fluorescent liposomes without a labeled enzyme or the existence of multiple enzymes within a single liposome, can be identified and omitted from subsequent smFRET analysis. Improved fluorophores for smFRET (brightness, photostability, monoexponential fluorescence lifetime, restricted blinking, and spectral fluctuations) will enable the recording of appropriate large numbers of FRET levels in order to corroborate the differing dwell times for the FRET levels, or the asymmetric distribution of rotor subunit orientations, respectively, in active and non-active F-ATP synthases.

### 4.4. Asymmetry in c-Ring Rotation

Cryo-EM structures of EcF_O_F_1_ exist in three states (Figure 6) that differ by the asymmetric positions of subunits γ and ε in relation to the peripheral stalk [12]. These structures likely correspond to the three 120° rotary positions of the catalytic dwells. However, the rotary positions of the **c**_10_-ring relative to subunit **a** in the three states were 108°, 108°, and 144° apart. These rotary positions are consistent with the mismatch between the orientation of the three catalytic sites in the EcF_1_ domain and the orientation of the **c**_10_-ring in EcF_O_ domain. The structure implies that the synthesis/hydrolysis of one particular ATP requires four **c** subunit steps (4H^+^ × 36° = 144°), while for each of the other two ATPs synthesized/hydrolyzed, three proton translocation-dependent single **c** subunit steps (3H^+^ × 36° = 108°) must occur (Figure 7). The three cryo-EM structures of the *Bacillus* PS3 F-ATP synthase show a similar asymmetry for the rotational states of the **c**-ring [29].

As a result of the asymmetric binding of subunits γ and ε to the **c**-ring, each **c** subunit is unique, such that residue **a**R210 is positioned between two adjacent **c**D61 residues of adjacent **c** subunit chains designated M and N in state 1 (PDB-id: 5T4O), chains S and T in state 2 (PDB-id: 5T4P), and chains P and Q in state 3 (PDB-id: 5T4Q). The three successive 120° ATP hydrolysis-driven power strokes rotate the **c**-ring between the three catalytic dwell positions that correspond to the three structures. During power stroke 1, **c** subunits M, V, U, T rotate past subunit **a**, while in power stroke 2 and power stroke 3, **c** subunits S, R, Q, and subsequently P, O, N rotate past subunit **a**, respectively. These power strokes are unique from each other, first because power stroke 1 translocates 4H^+^, while the other two each translocate 3H^+^. Second, a portion of subunit γ that extends beyond the diameter of the **c**-ring must pass through the narrow gap created by the peripheral stalk during power stroke 1. A portion of subunit ε that also extends beyond the **c**-ring diameter must pass through this gap during power stroke 3, whereas power stroke 2 is not affected in this manner (Figure 6). However, from the static cryo-EM structures, it is unclear how **c**_10_-ring rotation might be affected. Third, the position of **a**R210 relative to the **c**D61 residues of the adjacent **c** subunits varies significantly among the three cryo-EM states. In structure 5T4Q, which serves as a reference structure for the 0° rotational position in our model shown in Figure 7, **a**R210 is extremely close to **c**D61 of chain P (4.5 Å). This state 3, that is formed after power stroke 2 and precedes power stroke 3, is equally separated from the other two states by the rotation of three **c** subunits (108°). However, in the 4 **c** subunit, power stroke 1 from state 1 to state 2, represented by structures 5T4O and 5T4P, respectively, there is a symmetry mismatch between the 144° **c**-ring rotation in F_O_ (i.e., the angle between the two green lines) and the 120° F_1_ rotation (i.e., the angle between the two red lines). In state 1, the rotary position of the **c**-ring relative to **a**R210 is closer to a green line and F_O_ dominates the formation of a dwell position, while in state 2 this distance is closer to a red line and F_1_ dominates the formation of a dwell position.

We proposed that the high, medium, and low probabilities of transient dwell formation (Figure 5c) depend on the rotary position of the ATP hydrolysis-dependent power stroke relative to the peripheral stalk [108]. It is likely that power stroke 3 represents the data sets with a high probability to form transient dwells, i.e., there is a high probability for subunit **a** to stop F_1_-ATP hydrolysis-driven CCW rotation or even to push the **c**-ring CW after a 120° rotation. This is because structure 5T4O is in a rotary position comparable to that of a transient dwell after a 108° rotation of the **c**-ring (green line in Figure 7), instead of a 120° rotation (red line in Figure 7) as induced by the catalytic events in F_1_. In contrast, power stroke 1 may be responsible for the data sets with a low probability of transient dwell formation, because in the structure 5T4P, the **c**-ring is in a position in line with a 120° rotation induced by F_1_ (red line), where a transient dwell has not been formed. In the third case, transient dwells may be formed before power stroke 2, as in the structure 5T4P residue D61 of the **c** subunit closest to **a**R210 is not in an optimal position to receive a proton, in comparison with structure 5T4Q, which can be compensated by thermal fluctuation. The drag imposed on power strokes 1 and 3, when the lobes of subunit γ and ε pass through the narrow gap created by the peripheral stalk, respectively, may also contribute to the probability of forming transient dwells during power strokes 1 and 3.

In contrast, measurements of the spring constant of the central stalk elements and smFRET data [59,134] revealed that the **c**-ring is elastically coupled to the γε-shaft, with the softest part being the interface between F_O_ and F_1_ [58]. Elastic power transmission between F_O_ and F_1_ is a central element in the coupling of the two nanomotors and allows the enzyme to run with high kinetic efficiency and with the same mechanism even if the number of **c** subunits varies between species [57]. The three structures in Figure 6 and Figure 7 represent three energy-minimized snapshots of the subunit ε inhibited enzyme. These structures occur only transiently in the active enzyme. For the chloroplast enzyme it was shown that F_O_ alone can rotate 10× faster than F_1_ [160,161], i.e., during ATP synthesis it can run ahead of F_1_ and build up and store elastic energy in the central shaft, depending on the stochastic binding of protons to **c**D61 in this Brownian ratchet. Accordingly, the orientation of **c**D61 towards **a**R210 during the same power stroke can vary in the active enzyme. Nevertheless, it is within the scope of this model that the step size is not always 120° but can be as small as 108° or as large as 144°.

## 5. Conclusions

Our results on the rotation of single F_O_F_1_ ATP synthases suggest that its asymmetric structure is reflected in its rotational dynamics and breaks rotational symmetry. This is corroborated by three different single-molecule techniques applied here, and by the fact that other research groups studying the ATP synthase from different organisms and with different techniques found similar static and dynamic behavior. Table 2 summarizes the experimental differences found for the *E. coli* F_O_F_1_ ATP synthase.

We found that during ATP hydrolysis in the F_1_ domain one dwell time is by a factor 1.5 longer than the other two in rotation experiments with actin filaments. In contrast, the results of smFRET experiments indicate that one dwell time is shorter than the other two by a factor of 1.4 or 1.6 during ATP hydrolysis or synthesis, respectively. When the donor fluorophore was placed at a different position, this factor was even smaller (1.3). In addition, our experiments with gold nanorods suggests that there is rotational asymmetry also present in the F_O_ domain, as the two half-channels in subunit **a** seem to have different accessibilities for protons. At this state, the origin of the different patterns is unknown, because it is unclear which structural elements contribute mostly to this asymmetry. It could be that the effects of the asymmetric elements vary due to the elastic flexibility of the enzyme. One also has to consider that it is not clear how these differences are, at least partially, caused by the short observation times in the smFRET experiments or the drag imposed by the actin filament or any other probe that is coupled to an immobilized enzyme complex. Yet, overall the differences in dwell times were small and do not impede the high kinetic efficiency (i.e., the high turnover rate under load) of the enzyme [57,162].

In conclusion, the maximum difference between the dwell times for three rotor orientations was less than 1.6. This shows that the overall effect of the asymmetric structural features on the rotational dynamics and dwell times is small. Of the asymmetric elements discussed, the ones that involve the **b**_2_ homodimer are the most likely candidates to prolong some, but not all, of the catalytic dwells, because the **b**_2_ homodimer is asymmetrically placed on one side of the (αβ)_3_-ring. Together, these subunits form a stiff element with a torsional stiffness of κ ≈ 700 pNnm [35]. On the other hand, the central rotor stalk itself is designed to store elastic energy during power transmission between the F_O_ and F_1_ parts [57,59,134,163]. The interface between F_1_ and F_O_, involving subunits γ, ε, and the **c**-ring, respectively, is at least one order of magnitude more elastic than the stator (κ ≈ 20 pNnm). The flexible lever on subunit β adds a similar amount of elastic flexibility, reducing the effective spring constant of the active enzyme to about 35 pNnm [35,58]. Thus, transient flexibility in parts of the enzyme allows reducing kinetic limping. It remains for future experiments to investigate the details and effects of the asymmetric elements on the catalytic activity of the different types of ATP synthases.

## Figures and Tables

**Figure 1 molecules-24-00504-f001:**
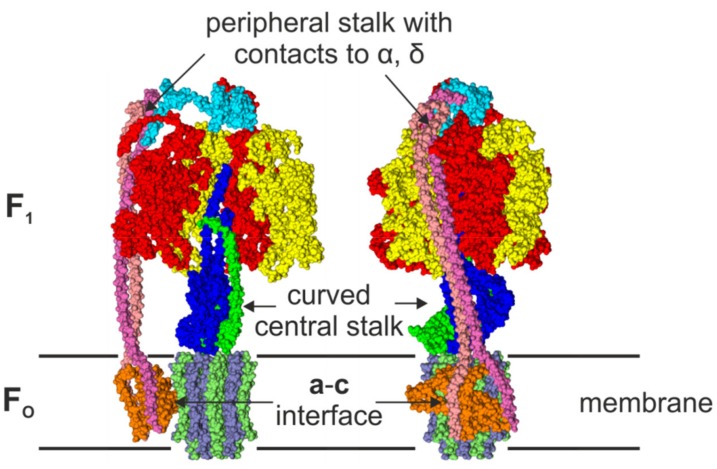
Front view (**left**) and side view (**right**) of a surface representation of the cryo-EM EcF_O_F_1_ structure (PDB-id: 5T4O) [12] in the membrane. One αβ-pair is omitted in the front view to reveal the conformation of the central stalk. The structure shows the asymmetric features, i.e., the peripheral stalk that is connected to subunit δ and one subunit α, the interface of the **c**-ring and subunit **a** with its two half-channels, and the curved central shaft composed of subunits γε. The subunits are colored in red (α), yellow (β), blue (γ), cyan (δ) green (ε), orange (**a**), pink, and mauve (**b**_2_), and ice blue/lime (**c**-ring).

**Figure 2 molecules-24-00504-f002:**
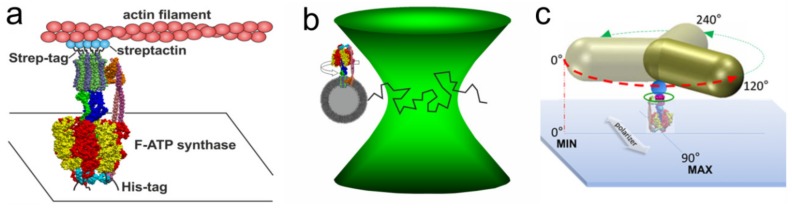
Single-molecule experiments with EcF_O_F_1_. (**a**) EcF_O_F_1_ was attached via His-tags in each β subunit to a cover glass. A fluorescently labelled actin filament, which was attached via streptactin and Strep-tags to the **c**-ring, served as a reporter for rotation during ATP hydrolysis [58]. (**b**) Fluorescently labelled EcF_O_F_1_ reconstituted in a liposome was diffusing through the confocal laser focus in a smFRET setup. Rotation of the central shaft during ATP hydrolysis was observed after addition of ATP, or ATP synthesis was powered by pmf [130]. (**c**) Immobilized EcF_O_F_1_ with an attached AuNR is rotating in 120° steps at saturating ATP concentrations, and intensities of the scattered light from the nanorod that were observed through a polarizer progressed sinusoidal and reached a maximum after 90° rotation [108].

**Figure 3 molecules-24-00504-f003:**
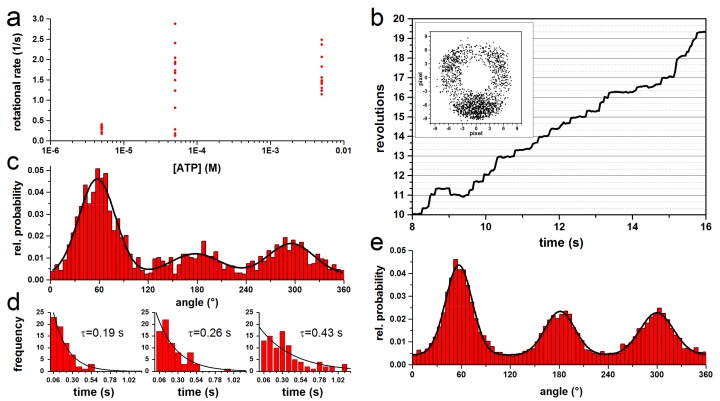
Properties of rotating actin filament-EcF_O_F_1_ complexes during ATP hydrolysis. (**a**) Rotational rate versus ATP. For short actin filaments (≤1 µm), the rotational rate is independent of the ATP concentration and the filament length. (**b**) Trajectory of a typical rotating complex at 100 µM ATP in steps of 120°. The inset shows the endpoints of the 0.8 µm actin filament. (**c**) Normalized histogram of the angular probability distribution of the complex in **b**, where the three peaks, separated by 120°, were fitted with a Gaussian. (**d**) Dwell time histograms of each peak in c fitted with monoexponential decay. The dwell times (τ) are given for each histogram. (**e**) Combined normalized histogram of the angular distribution of rotating actin filament from 13 single molecules at 50–5000 µM ATP with 513 to 2174 frames per molecule. Each dataset was fitted with three Gaussians to determine the area of each peak. The datasets were aligned by positioning the largest peak at 60°.

**Figure 4 molecules-24-00504-f004:**
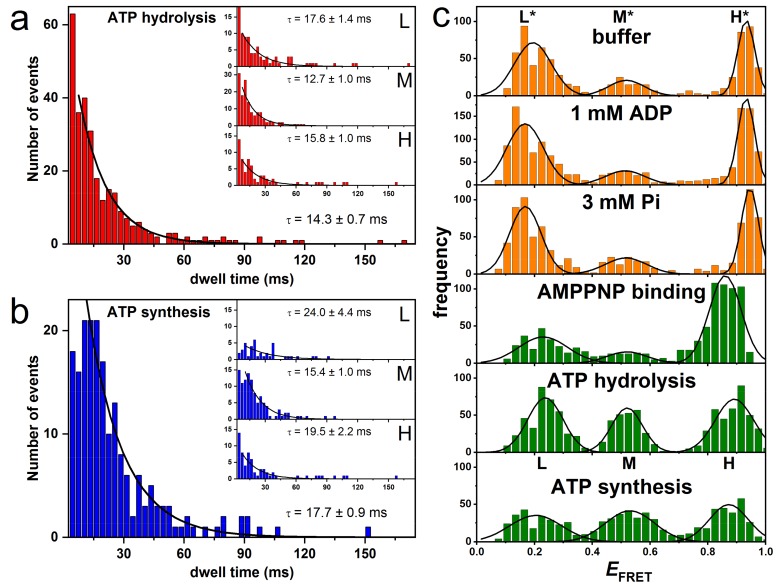
FRET histograms and dwell time histograms of subunit ε rotation in single FRET-labeled EcF_O_F_1_ during catalysis. (**a**) Dwell time histogram for all FRET levels during ATP hydrolysis. The insets show the individual dwell times for each of the three FRET levels. (**b**) Dwell time histogram for all FRET levels during ATP synthesis. The insets show the individual dwell times for each of the three FRET levels. (**c**) FRET histograms showing the three FRET levels L*, M*, H* (orange), and levels L, M, H (green) from measurements at different conditions at pH 8. See text for details. (Modified from [130]).

**Figure 5 molecules-24-00504-f005:**
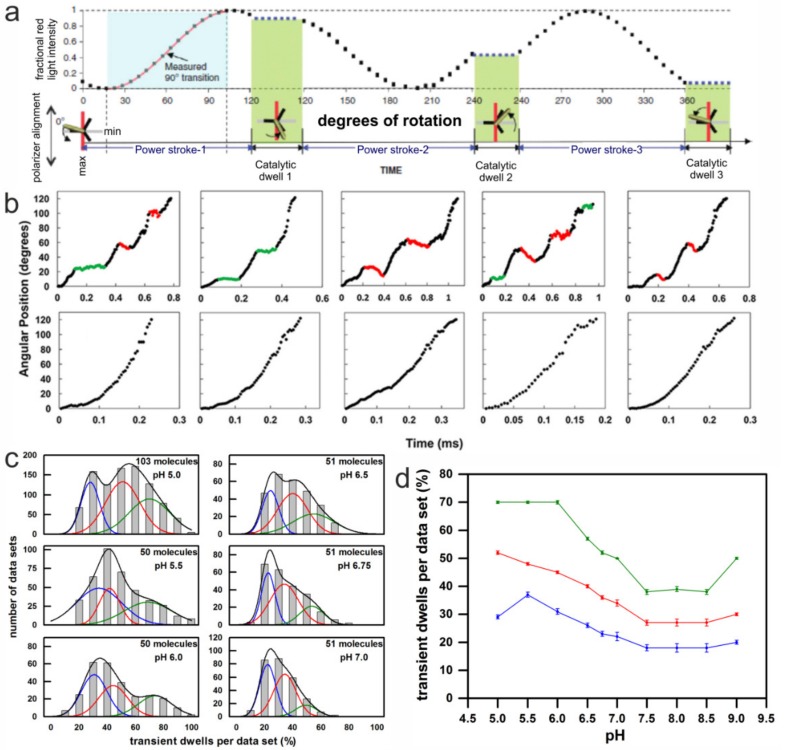
Method and results of single-molecule rotation experiments with AuNR attached to the **c**-ring of immobilized EcF_O_F_1_ during ATP hydrolysis. (**a**) Theoretical plot of the intensity of scattered red light from a nanorod during one complete revolution that involves three consecutive power strokes and three consecutive catalytic dwells separated by exactly 120°. The nanorod is initially positioned almost—but not exactly—perpendicular to the orientation of the polarizer, such that the scattered light intensity goes through a minimum then a maximum prior to catalytic dwell 1. A transition includes the data between the minimum and maximum intensities representing 90° of the 120° of rotation. For analysis transitions from power stroke-1 to dwell 1 were selected. (Modified from [56]) (**b**) Examples of time-dependent changes in rotational position of n-EcF_O_F_1_ during F_1_-ATPase–dependent power strokes at pH 5.0, where transient dwells, either present (**top**) or absent (**bottom**), are shown. Transient dwells where rotation was halted or contained CW rotation are colored green and red, respectively. (**c**) The distribution of single-molecule n-EcF_O_F_1_ power stroke data sets as a function of the percentage of the occurrence of transient dwells per data set, which were binned to each 10% (gray bar graphs). Each data set contained ~300 power strokes, derived from the indicated number of molecules. The data were fitted to the sum of three Gaussians (black line), where the probability of forming transient dwells was low (blue line), medium (red line), and high (green line). (**d**) The pH dependencies of the average percent of transient dwells per data set, with standard errors derived from the three Gaussians. (Modified from [108]).

**Figure 6 molecules-24-00504-f006:**
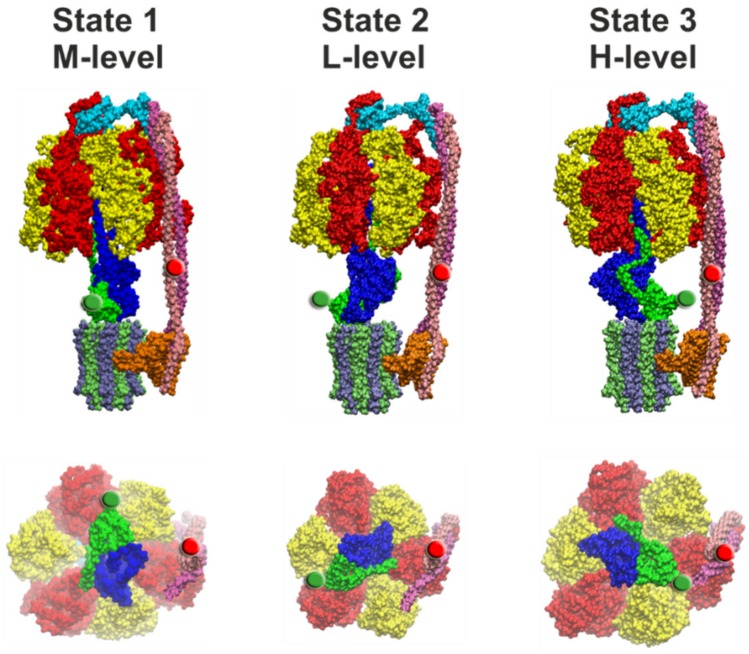
Side view (top row) and view from the membrane side with the **c**-ring and subunit **a** omitted of the three cryo-EM structural states of EcF_O_F_1_ [12]. State 1 (PDB-id: 5T4O, left), state 2 (PDB-id: 5T4P, middle), and state 3 (PDB-id: 5T4Q, right) correspond to FRET levels M, L, and H, respectively (as in Figure 4). In each image, the FRET-donor position at ε56 is marked with a green sphere and the FRET-acceptor position at **b**64 is marked with a red sphere. The portions of subunit ε and γ that extends beyond the diameter of the **c**-ring must pass through the narrow gap created by the peripheral stalk during rotation from state 3 to state 1, and during rotation from state 1 to state 2, respectively. The subunit coloring is the same as in Figure 1.

**Figure 7 molecules-24-00504-f007:**
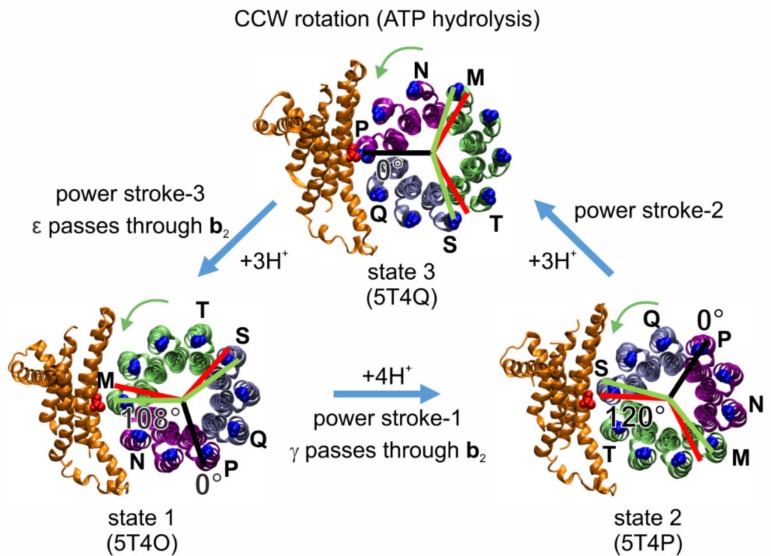
Asymmetry between the three F_1_-ATPase power strokes of EcF_O_F_1_, as shown by the rotary positions of residue **c**D61 in each **c** subunit (blue) relative to **a**R210 (red) in the three states of the EcF_O_F_1_ cryo-EM structures (viewed from the membrane side as in Figure 6) designated by their PDB-ids [12]. In structure 5T4Q, **c**D61 of **c** subunit chain P is closely aligned with **a**R210 (black line designate 0° of rotation). Assuming that this state shows the alignment with a catalytic dwell position in F_1_, red lines show the other two 120° rotational positions of the catalytic dwells in F_1_ during subsequent power strokes. Green lines show the 108° rotations that result from the three proton-dependent CCW rotational steps (green arrows) of the **c**-ring that occur during the power strokes, which results in strain between the catalytic dwell positions of F_O_ and F_1_. From state 3 to state 1 (3 **c** subunits, purple), and from state 2 to state 3, the **c**-ring rotates by 108° (3 **c** subunits, ice blue), whereas from state 1 to state 2, the **c**-ring rotates by 144° (4 **c** subunits, lime).

**Table 1 molecules-24-00504-t001:** Distribution of the three different states/rotor orientations of the F/A/V-type complexes that were found in cryo-EM studies. The F_O_F_1_ structures were aligned according to the orientation of the central stalk in relation to the peripheral stalk. As A_O_A_1_ and V_O_V_1_ have two and three peripheral stalks, respectively, it is not possible to align their states with the F_O_F_1_ structures.

Structure	State 1	State 2	State 3	Nucleotide Occupancy	Reference
EcF_O_F_1_	46.3%	30.0%	23.7%	ADP in αβ_closed_	[12]
BPF_O_F_1_	45.3%	35.1%	19.6%	Mg-ADP in αβ_TP_, Pi in αβ_E_	[29]
CF_O_F_1_	8.1%	7.5%	84.4%	Mg-ADP in αβ_DP_ and αβ_TP_	[27]
MF_O_F_1_	31.3%	35.0%	33.7%	Unknown	[24]
TtA_O_A_1_	72.5%	18.7%	8.8%	ADP in AB_closed_ and AB_semi-closed_	[96]
YV_O_V_1_	47%	36%	17%	unknown	[94]

**Table 2 molecules-24-00504-t002:** Dynamic and static differences of measurements with EcF_O_F_1_. The top part shows the dynamic differences in dwell times for the three catalytic dwells found in our single-molecule fluorescence experiments. For comparison, the bottom row shows the static distribution of the three different states in the cryo-EM measurement. (* FRET-levels do not apply to this method.)

FRET-Level Method	M, Dwell Time 1 (ms)	L, Dwell Time 2 (ms)	H, Dwell Time 3 (ms)	Direction of Rotation
rotation assay^*^	300.0	200.0	200.0	ATP hydrolysis
smFRET ε56/**b**64	12.7	17.6	15.8	ATP hydrolysis
smFRET ε56/**b**64	15.4	24.0	19.5	ATP synthesis
**Conformation**	**State 1**	**State 2**	**State 3**	
cryo-EM^*^	47%	30%	24%	N.A.

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
