# Peer review of "Structural Asymmetry and Kinetic Limping of Single Rotary F-ATP Synthases"

_molecules, 2019, doi:10.3390/molecules24030504_

Reviewer 1 Report

This is an exciting review article on rotary ATP synthase written by a group of experts. This review will help students/postdoctoral fellows and scientists of related field to understand the unique enzyme.  Their proposed rotation mechanism is still difficult to follow. I suggest authors to write more elaborately. Unfortunately, we have no time to discuss in detail, since the editors gave not enough time for us. Using the schemes for the kinetic steps combined with rotation angles of a rotor will help readers to follow. Furthermore, the extensive editing and elaboration of manuscript are required before it is accepted for publication. I have discussed the examples where elaboration and editing are required

(1) Since the title and Abstract should be self-explanatory, “kinetic limping” (title) and “one particular dwell” (Abstract) should be elaborated, or use other words. 

(2) I found many mistakes in writing in this manuscript. Thus, all authors should read the manuscript and revise it extensively. I am giving you several examples for the first three pages. Authors should check the entire manuscript extensively.

1) line 31: “on the expense of the proton motive force (pmf) over the membrane” should be “coupling with the proton motive force established across the membrane”.

2)line 37: “The structure” should be elaborated. 

3) line 39: “……..that was followed was by……….”  Is a strange sentence, and should be revised.

4) Using different nomenclatures for one enzyme is confusing: we found FoF1 ATP synthase, F-ATPase, EcFoF1, F-type ATP synthase, and F1 ATP synthase. Similarly, they should use either F1 or EcF1 and Fo or EcFo.

5)  line 40:“complete ATP synthase”;  “complete” should be omitted.

6)  line 52: What is “its” ?

7)  line 69: “positive (P) or the negative (N) side of the membrane” should be elaborated.

8)  line 70-71: “hydrophobic core of the membrane” should be elaborated.

9)   line 72: “This reflects an adaptation to their environment.”: Does this sentence can be included here. 

10)  line 82: “the F-ATP synthase as, I” should be corrected.

 11)   line 96: “Protons from the P-side”: what is P-side?

12)  line 102: define CCW.

13) line 106: “but not in A/V-type enzymes, where the two dwells are at the same rotor position”: incomplete sentence. 

14)   Paper on rotation experiments on yeast vacuolar ATPase is not cited:Hirata,et al.  J. Biol Chem. 278, 23714. This is the first paper reported rotation of yeast V-ATPase.

Author Response

Reviewer 1:

This is an exciting review article on rotary ATP synthase written by a group of experts. This review will help students/postdoctoral fellows and scientists of related field to understand the unique enzyme.  Their proposed rotation mechanism is still difficult to follow. I suggest authors to write more elaborately. Unfortunately, we have no time to discuss in detail, since the editors gave not enough time for us. Using the schemes for the kinetic steps combined with rotation angles of a rotor will help readers to follow. Furthermore, the extensive editing and elaboration of manuscript are required before it is accepted for publication. I have discussed the examples where elaboration and editing are required.

 (1) Since the title and Abstract should be self-explanatory, “kinetic limping” (title) and “one particular dwell” (Abstract) should be elaborated, or use other words. 

We have maintained our title because the term "kinetic limping" has been introduced previously to describe the asymmetric kinetics in the ATP-driven molecular motor kinesin (for example A. N. Fehr et al., "On the Origin of Kinesin Limping", Biophys J 97, 1663, 2009).

(2) I found many mistakes in writing in this manuscript. Thus, all authors should read the manuscript and revise it extensively. I am giving you several examples for the first three pages. Authors should check the entire manuscript extensively.

1) line 31: “on the expense of the proton motive force (pmf) over the membrane” should be “coupling with the proton motive force established across the membrane”.

We have changed the sentence to “ATP synthesis is coupled to the proton motive force (pmf) across the membrane.”

2) line 37: “The structure” should be elaborated. 

We have specified the misleading term and changed it to "Structural details of the EcFO domain were revealed by NMR... ".

3) line 39: “...that was followed was b...”  Is a strange sentence, and should be revised.

We are sorry for this mistake and amended it.

4) Using different nomenclatures for one enzyme is confusing: we found FoF1 ATP synthase, F-ATPase, EcFoF1, F-type ATP synthase, and F1 ATP synthase. Similarly, they should use either F1 or EcF1 and Fo or EcFo.

We apologize for the confusion. We have amended this. Throughout the text we now use the following terms:

-       F-ATP synthase, A-ATP synthase, V-ATPase

-       F/A/V-type complexes in cases when we refer to ATP synthases and ATPases at the same time.

-       F1 or FO in cases when we refer to a common feature of all F-ATP synthases.

-       xF1 or xFO in cases when we refer to a specific feature of F-ATP synthases from a specific species x.

5)  line 40:“complete ATP synthase”;  “complete” should be omitted.

We thank the reviewers for pointing this out. We have removed the word complete in all cases.

6)  line 52: What is “its” ?

‘Its’ refers to the symmetry of the central stalk. We have clarified this in the text.

7)  line 69: “positive (P) or the negative (N) side of the membrane” should be elaborated.

We are sorry that we did not make this clear. The P-side and N-side refers to the positive and negatively charged side of a membrane. We have clarified this in the text.

8)  line 70-71: “hydrophobic core of the membrane” should be elaborated.

We have changed this  phrase to “located in the middle of the hydrophobic membrane.”

9)   line 72: “This reflects an adaptation to their environment.”: Does this sentence can be included here. 

We think it can be included, because it gives a reason for the different numbers of c subunits found in different species. We combined it with the previous sentence.

10)  line 82: “the F-ATP synthase as, I” should be corrected.

We thank the reviewer for pointing out this mistake, which we have corrected.

11)   line 96: “Protons from the P-side”: what is P-side?

See point 7)

12)  line 102: define CCW.

CCW means counterclockwise and is defined in line 84.

13) line 106: “but not in A/V-type enzymes, where the two dwells are at the same rotor position”: incomplete sentence. 

We thank the reviewer for pointing out this mistake, which we have corrected.

14)   Paper on rotation experiments on yeast vacuolar ATPase is not cited:Hirata,et al.  J. Biol Chem. 278, 23714. This is the first paper reported rotation of yeast V-ATPase.

Thank you for drawing our attention to this publication. We have added the reference.

We have thoroughly read the manuscript again and we hope to have corrected all mistakes and clarified all unclear passages to the reviewers satisfaction.

All changes in the text have been marked in 'red'.

Reviewer 2 Report

In their manuscript, Sielaff et al. review recent work on the non symmetrical action of the rotary motor FOF1 ATP synthase. The manuscript covers the important findings in a well-structured format. It represents a valuable contribution to the field of motor proteins.

Their manuscript also seems to include original work, which is as a matter of fact not clearly differentiated. This should be improved revising the manuscript. In table and figure captions references should be given where applicable.

The authors mention several times the unexpectedly small dwell time differences without reasoning why those differences are expected to be larger. This should be worked out.

Further hints:

Line 299: The number of digits for the errors exceeds that of the values themselves, this should be corrected.

Typos:

Line 453 viscous instead of vicious

Line 471 symmetry instead of symmetricni

Line 550 FRET level instead of FREL level

Line 657 hexamer instead of heaxamer

The plural of level is levels, this was ignored on several occurrences

Author Response

Reviewer 2:

In their manuscript, Sielaff et al. review recent work on the non symmetrical action of the rotary motor FOF1 ATP synthase. The manuscript covers the important findings in a well-structured format. It represents a valuable contribution to the field of motor proteins.

Their manuscript also seems to include original work, which is as a matter of fact not clearly differentiated. This should be improved revising the manuscript.

We thank the reviewer for this comment. For our analysis we have reevaluated our previously published data and some unpublished data to search for persistent asymmetric patterns. We have clarified this in the text.

In table and figure captions references should be given where applicable.

The respective references for each figure were given in the manuscript already, therefore we did not add additional references there.

The authors mention several times the unexpectedly small dwell time differences without reasoning why those differences are expected to be larger. This should be worked out.

We are sorry for this confusion. We believe that there is no strong evidence why this differences should be larger. We have deleted the word ‘unexpectedly’ in all cases.

Further hints:

Line 299: The number of digits for the errors exceeds that of the values themselves, this should be corrected.

We changed 0.3 ± 0.21 s to 0.3 ± 0.2 s, but did not change 0.2 ± 0.15 s, and 0.2 ± 0.12 s, because the errors start with the digit ‘1’, in which case it is advised to keep the digit after ‘1’.

Typos:

Line 453 viscous instead of vicious

Line 471 symmetry instead of symmetricni

Line 550 FRET level instead of FREL level

Line 657 hexamer instead of heaxamer

The plural of level is levels, this was ignored on several occurrences

We thank the reviewer for carefully pointing out these mistakes. We have corrected all typos according to the reviewers advice.

All changes in the text have been marked in red.

Reviewer 3 Report

This is an interesting and well-written review of the author’s work on single rotary E. Coli F0F1 ATP synthases. The authors discussed specifically the relationship between structural and rotational symmetry/asymmetry associated with the enzyme’s rotary progression during ATP hydrolysis using three single-molecule fluorescence microscopy techniques. A few typos should be revised before publication:

Line 39: “that was followed was by” should be revised.

Line 82: “as,i” should be removed?  

Line 104: c ring should be c-ring

Line 106: There is a sentence fragment and should be combined.

Author Response

Reviewer 3:

This is an interesting and well-written review of the author’s work on single rotary E. Coli F0F1 ATP synthases. The authors discussed specifically the relationship between structural and rotational symmetry/asymmetry associated with the enzyme’s rotary progression during ATP hydrolysis using three single-molecule fluorescence microscopy techniques. A few typos should be revised before publication:

 Line 39: “that was followed was by” should be revised.

Line 82: “as,i” should be removed?  

Line 104: c ring should be c-ring

Line 106: There is a sentence fragment and should be combined.

We thank the reviewer for pointing out these mistakes. We have corrected all typos and the misplaced sentence fragment has been incorporated into the main text. We have marked all text passages that we have changed in 'red'.

Round  2

Reviewer 1 Report

As authors have revised the manuscript extensively, it is now acceptable for publication.